# Study on the Stress Distribution and Stability Control of Surrounding Rock of Reserved Roadway with Hard Roof

Yuxi Hao [1], Mingliang Li [1], Wen Wang [2,*], Zhizeng Zhang [1] and Zhun Li [1]

[1] School of Architectural Engineering, Zhongyuan University of Technology, Zhengzhou 450007, China; 6548@zut.edu.cn (Y.H.); 2021109363@zut.edu.cn (M.L.); 6175@zut.edu.cn (Z.Z.); 2022109405@zut.edu.cn (Z.L.)
[2] College of Energy Science and Engineering, Henan Polytechnic University, Jiaozuo 454000, China
* Correspondence: wangwen2006@hpu.edu.cn; Tel.: +86-0391-3987973

**Abstract:** According to field observation and theoretical analysis, the failure of the 1523103 reserved roadway is mainly affected by the lateral support pressure, rock mass strength, and support mode. With the mining of the 152309 working face, the lateral pressure of coal pillars on both sides of the reserved roadway increases, and since the lithology of the two sides and the floor of the roadway is weak, the reserved roadway experiences spalling and floor heave. Through numerical simulation, the distribution law of surrounding rock stress and the displacement of surrounding rock are obtained after the roof cutting and pressure relief of the reserved roadway with hard roof. According to the cause of surrounding rock failure of a reserved roadway, the combined control technology of roof cutting and pressure relief, grouting anchor cable support, and bolt support is put forward. After cutting the roof and releasing the pressure on the working face, the lateral support pressure of the two sides of the roadway is significantly reduced, the deformation of the two sides of the roadway is small, the maximum shrinkage rate of the section is reduced from 70% to 11%, and the deformation of the surrounding rock of the 1523103 reserved roadway is effectively controlled. The successful control of the surrounding rock in the 1523103 tunnel reduces the number of coal pillars to be installed, improves the coal extraction rate, and is conducive to the sustainable utilization of limited natural resources and the sustainable development of the coal industry.

**Keywords:** reserved roadway; roof cutting and pressure releasing; surrounding rock damage; hard roof; grouting anchor

## 1. Introduction

In the field of coal mine safety prevention and control, the control process of the surrounding rock of a reserved roadway is a complex process that runs through each deformation stage of the surrounding rock [1], which has an important impact on the sustainable development of the coal mining industry. The sustainable development of the coal mining industry needs to realize the effective utilization of coal resources, the effective protection of coal miners, and the effective protection of the environment through scientific technology and reasonable management methods. This requires us to analyze the main factors causing roadway deformation according to the basic principle of roadway surrounding rock deformation control and put forward a reasonable surrounding rock deformation control scheme under the premise of sustainable development of the environment and industry. In response to such problems, many scholars have studied this phenomenon in an attempt to uncover solutions.

In recent years, coal mine resources have been depleted and the coal mining rate is low, which is similar to the problems faced by the construction industry as analyzed in the article by Zhimin Wang [2] and others. These phenomena prompt us to transform traditional technology into intelligent technology to extend the service life of mines. The traditional roadway surrounding rock control process is complicated, and the three-dimensional platform construction of BIM [3] technology can achieve three-dimensional visualization of

mine construction through digital geological models, realize roadway safety monitoring, and improve the efficiency and effect of construction management. Feng Wan [4] used roof cutting and pressure relief combined with concrete walls to control the deformation of surrounding rock in the tunnel, and Jianjun Ma [5] used a dynamic load concrete thermoelastoplastic damage coupling model to study the dynamic load effect of concrete. In terms of the numerical simulation of tunnel surrounding rock, Jianjun Ma [6] used a fast particle accumulation generation algorithm with discontinuous deformation and controllable gradient to simulate the mechanical properties of the rock. Regarding the impact of advanced blasting on surrounding rock, Jianjun Ma [7] applied the M-JHB4DLSM model to study the mechanical behavior of tunnels under close-range blasting. In recent years, Academician Hongpu Kang, in view of the supporting problems such as the large deformation of surrounding rock of kilometer-deep roadways, and on the basis of studying the deformation and failure mechanism of the surrounding rock of roadways, put forward the cooperative control theory of roadway surrounding rock support + modification + pressure relief "trinity". A complete set of technology systems for the control of surrounding rock of deep roadways was developed, which integrates the active support of high prestressed anchor cable, active modification of high-pressure fracturing grouting, and active pressure relief via hydraulic fracturing. The new material of NPR anchor steel, developed by Academician Manchao He, has the characteristics of being non-magnetic, with high strength, high toughness, and high uniform extension, which solves the local deformation and fracture of ordinary anchors and is the first kind of material with these characteristics at home and abroad. Up to now, in the field of tunnel surrounding rock control, there are mainly five surrounding rock control solutions: surface support technology, anchoring technology, improving the self-supporting capacity of surrounding rock, pressure relief technology, and joint control technology. In order to adapt to various construction environments, joint control technology is widely used. Joint control technology can give full play to the advantages of various support methods and achieve complementary advantages. In 2008, He [8–12] proposed the theory of "Roof cutting short-wall beam" via field investigation and indoor experimental research and formed the "longwall mining 110 method" on this basis. Then, by comparing the traditional gob-side entry retaining technology, a new technology involving roof cutting and pressure relief gob-side entry retaining was proposed. Roof cutting and pressure relief technology can not only reduce the stress of surrounding rock, ensure safety in the coal mining process, reduce the risk of accidents, and reduce environmental pollution, but also improve coal recovery and reduce waste so as to prolong the service life of coal resources. All of these advantages help to protect coal resources, provide more resources for the future, realize the effective use of resources, and promote the sustainable development of geotechnical engineering. He believes that the gangue that is produced after roof pre-splitting blasting can play a supporting role in the overlying strata due to its own fragmentation and expansion, so that the roof forms a short-arm beam structure in the lateral direction. And the filling mining method of disposing coal gangue in the goaf of the working face can not only reduce the discharge of solid waste in a coal mine, but also reduce the disaster of mining subsidence and improve the recovery rate of mine resources. It is one of the key technical ways to realize green mining in coal mines.

The fracture characteristics and activity laws of the overlying rock layer along the empty alley are closely related to the fault structure characteristics and activity laws of the overlying rock layer during the working face of the upper section and the working face of this section, but they also have their own characteristics and rules. Zhu et al. [13] analyzed the variation law of the horizontal stress distribution of hard and thick roof plates using a mechanical formula combined with a numerical simulation research method and obtained the basic top first pressure step formula, which achieved a better early warning effect and provided protection for the safety of miners as much as possible. Chen et al. [14] studied the parameters of the deep hole pre-cracked cutting roof of the hard roof and narrow coal column protection alley with respect to the excessive deformation of the surrounding rock of the hard roof roadway. Reducing the loss of coal pillar while ensuring safety, high yield,

and high efficiency is an important part of coal mining. The application of the technology of retaining a narrow coal pillar in pre-splitting blasting can effectively reduce the roof compression strength, reduce the gas concentration in the goaf, reduce the coal spontaneous combustion rate, effectively improve the recovery rate of coal resources, and promote the sustainable development of the coal industry. Wang et al. [15] studied the key parameters of the coal column combined with the existing theory and numerical simulation software and obtained the breaking position of the roof plate of the goaf area, and they believed that there was a trapezoidal internal stability zone around the coal column.

Due to the variability of surrounding rock properties, quarry stress, and roadway section size, the mechanism of roadway deformation and failure is diverse and complex, resulting in different methods and countermeasures for roadway support [16]. Different roadway support methods also have different effects on the sustainable development of coal mines. The traditional roadway support mainly uses a steel frame support and wood support, which have many problems, such as the fact that a steel frame support will produce a lot of scrap steel and a wood support will waste a lot of wood resources. At the same time, these support methods also have safety risks. Modern roadway support methods pay more attention to sustainable development, mainly using new materials and technologies such as bolt support, anchor wire mesh support, shotcrete support, and so on. These support methods can not only improve the stability and safety of the roadway, but also reduce the impact on the environment, save resources, and reduce costs, which is conducive to the sustainable development of coal mines. Qin et al. [17] analyzed the distribution characteristics of the surrounding rock-bearing structure of the roadway using the existing theory, used the FLAC3D 6.0 finite element software to perform a numerical simulation, and proposed a reinforcement scheme according to the analysis results, and the experimental results proved that the bearing capacity of the shallow surrounding rock of the roadway was significantly improved after the application of the new support scheme. Taking a coal mine as the research background, Yang [18] and others used the research method of field practice + theoretical analysis + numerical simulation to analyze the pressure relief effect of severe roadway deformation on surrounding rock, and verify the feasibility of using the pressure relief effect to excavate a new roadway near the damaged roadway, which reduces the waste of coal resources and improves the efficiency of coal mining. Wang et al. [19] took a coal mine in the western part of the Qingyang mining area as the engineering background, used the method of field practice + numerical calculation to reveal the distribution law of the stress field and the displacement field of the surrounding rock of each section, and optimized the roadway support parameters. Xie [20] proposed "external anchor-internal unloading" collaborative control technology according to the deformation of the surrounding rock of a certain mine, and the new technology achieved a good surrounding rock control effect in a field application. A steel arch + bolt support system is widely used in roadway support [21], but different types of bolts have different ways of connecting steel arches, and their supporting effects are also different [22]. Strengthening the retractable steel support with the steel strand bolt passing through the steel beam can maintain the stability of the roadway even under mining stress. However, this method requires the use of a large amount of steel, which will inevitably be wasted in the support process. According to the geological conditions of a steeply inclined coal seam in a coal mine, Tu [23] developed a similar simulation system for a steeply inclined coal seam. The simulation results prove the effectiveness of roof caving and gangue filling characteristics in the process of steeply inclined coal seam mining. Filling goaf with gangue can not only reduce the discharge of solid waste from a coal mine, but also reduce the disaster of mining subsidence and improve the recovery rate of mine resources. Taking the deformation problem of a deep-buried soft rock roadway as the background, Guo et al. [24] studied the causes of roadway deformation and the failure of the original support, and put forward a new full-section combined support system involving an anchor cable, mesh shotcreting, and U-shaped steel support. Shavarskyi et al. [25] studied the influence of longwall face advance on the stress–strain state of a rock mass using GeoDenamics, determined the

physical and geometric parameters of the reference pressure and the parameters of a lower sandstone filling body during the subsidence of an old roof, and analyzed the influence of additional pressure on a direct roof. In view of the surrounding rock deformation of a reserved roadway under the condition of a hard roof, most scholars study how to strengthen the surrounding rock, but there is little research on the pressure relief of the surrounding rock and the sustainable development of the environment and economy.

In this paper, by cutting the roof of the working face and blasting along the strike, the influence of the goaf on the roadway roof is reduced, the stress caused by the goaf roof to the roadway is cut off, and pressure relief is achieved. According to the failure reasons of the reserved roadway with a hard roof, the combined control technology of roof cutting and pressure relief, grouting anchor cable, and bolt support is put forward, which reduces the number of protected coal pillars and increases the coal mining rate, and can prolong the service life of the coal mine. It is beneficial to the sustainable utilization of non-renewable resources and the sustainable development of the coal industry, and can provide a reference for similar surrounding rock control work.

## 2. Project Overview

### 2.1. Geological Overview

The Gushuyuan Mine is located in Jincheng Mining Area, Shanxi, China, 1 km from Jincheng District. The No. 15 coal seam is the main coal mining seam, located in the lower part of the Taiyuan Formation. The coal seam burial depth is about 420 m; the mining width is 181 m; the mining length is 1588 m; the coal thickness is 0.81~3.56 m with an average of 2 m; and the coal seam inclination angle is 1~9° with an average of 5°. The average compressive strength of the coal seam was measured to be 22.68 MPa, which was relatively low. The direct roof of the No. 15 coal seam is made of K2 limestone, with a thickness of about 9.00 m and an average compressive strength of 92.41 MPa, which is relatively high. The floor is mainly composed of black-gray mudstone and bauxite mudstone. The average compressive strength is 46.88 MPa, and the strength is low. The maximum horizontal principal stress of the No. 15 coal seam is 9.45 MPa, the maximum–minimum horizontal principal stress is 7.11 MPa, and the lateral pressure coefficient is greater than 1, which is mainly horizontal stress and belongs to the type of tectonic stress field.

### 2.2. Overview of the Roadway

The No. 15 coal seam of the Gushuyuan Mine is the main coal mining seam, and the working face adopts the layout form of "two into one round", with the 1523092 alley and 1523103 alley as the inlet alleys, and the 1523091 alley as the return alley, as shown in Figure 1. The 1523091 roadway and the 1523092 roadway collapsed after mining. The 1523103 reserved roadway was retained during the mining process to serve the 152310 working face. However, due to the influence of the mining process, the surrounding rock of the 1523103 reserved roadway was serious. The failure depth is 4.54 m~5.96 m, and the maximum floor heave is 1.5 m. The roof of the roadway has almost no deformation but shows an overall sinking trend. The maximum section shrinkage rate is more than 70%, and the deformation is shown in Figure 2.

The section of the 1523103 roadway is designed as a rectangular roadway with dimensions of 5.6 m × 2.6 m. The original support method is a combined support of a bolt and anchor cable. One anchor cable with a length of 5400 mm and with row spacing of 800 mm is installed on the roof of the roadway, and two high-strength screw steel bolts without longitudinal reinforcement with a diameter of 18 mm and a length of 2000 mm are installed. The row spacing is 3600 mm × 800 mm. Three high-strength screw steel bolts without longitudinal reinforcement with a diameter of 18 mm and a length of 2000 mm are installed on each side, and the row spacing is 800 mm × 800 mm, as shown in Figure 3.

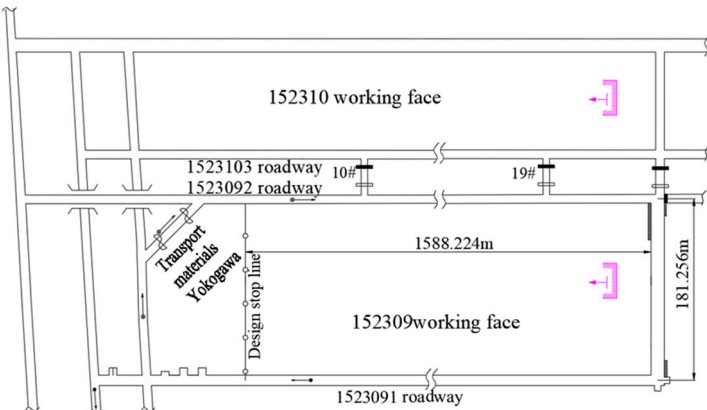

**Figure 1.** The layout diagram of reserved roadway and working face.

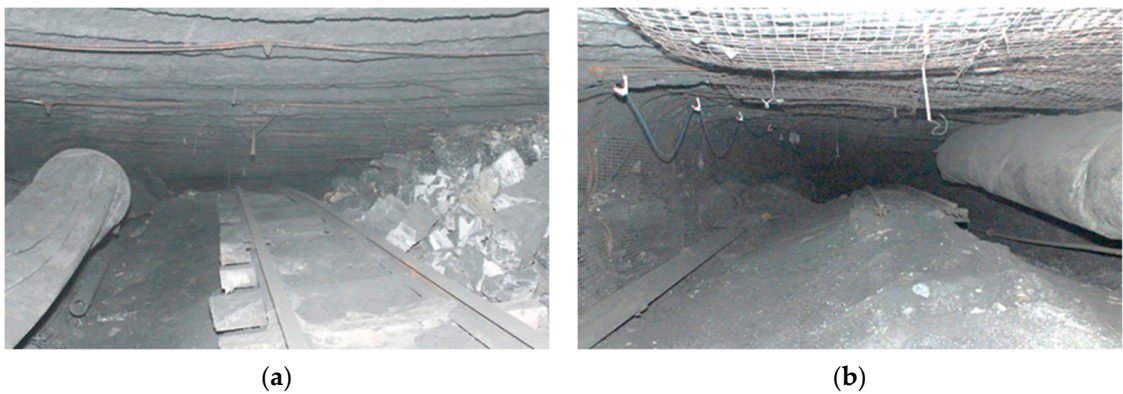

(**a**)                                                                          (**b**)

**Figure 2.** Real pictures of surrounding rock deformation of 1523103 roadway. (**a**) Roof subsidence; (**b**) bottom drum situation.

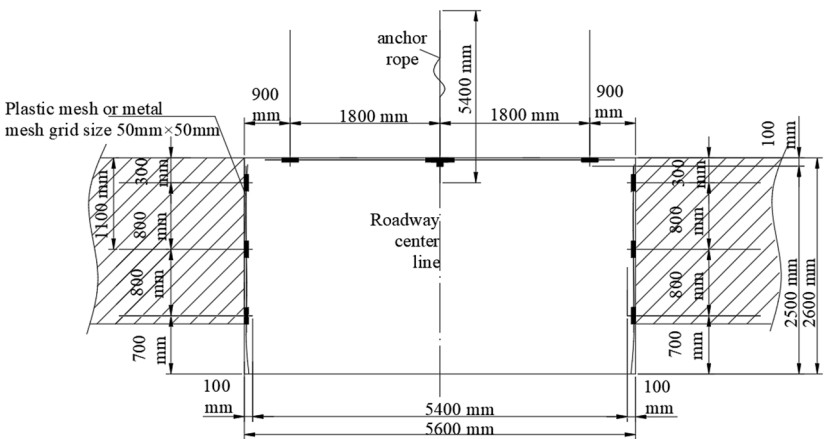

**Figure 3.** The original support method of 1523103 roadway.

## 3. Cause Analysis of Surrounding Rock Failure of Reserved Roadway

### 3.1. Justification of the Geomechanical Model

According to the actual production conditions of the 152309 working face, a 280 m × 280 m × 70 m model was established via FLAC3D. According to the actual situation, the dip angle of the roof and floor strata was set to 0°; the thickness of the coal seam mined by the model was 2 m; the length of the working face was 170 m; the size of the reserved roadway was 5 m × 2.5 m, measured from the bottom of the fixed model, the front and back, and the left and right; the advancing direction of the working face was measured from bottom to top; the reserved roadway was located at 25 m on the right side of the

working face; and the grid near the roadway was encrypted. An analysis of the statistical data of the measured vertical stress from all over the world shows that the vertical stress increases linearly in the range of depth of 25–2700 m, which is roughly equivalent to the gravity calculated according to the average bulk density. According to the data from the geological exploration report, the average bulk density of rock mass within 320 m below the surface is $2.4 \times 10^4$ N/m$^3$. The upper boundary of the model was 320 m from the surface, the vertical ground stress was 7.9 MPa, and the lateral pressure coefficient was 1.2. The model is shown in Figure 4.

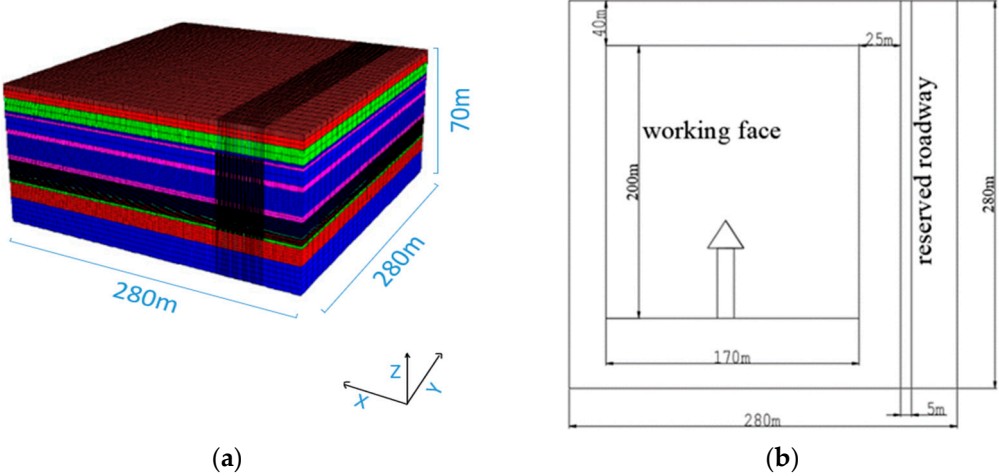

(**a**)　　　　　　　　　　　　　　　　(**b**)

**Figure 4.** Numerical model and working face and reserved roadway layout diagram. (**a**) Numerical model. (**b**) Working face and reserved roadway layout diagram.

The physical and mechanical properties of a coal–rock mass are affected by joints, cracks, and other structural planes, and sometimes, the influence of structure on the properties of a coal–rock mass is much greater than that of the material itself, so, in general, the properties of a coal–rock mass and a rock block are very different. In a numerical simulation, the mechanical parameters such as the elastic modulus, cohesion, and tensile strength of a rock mass are generally 1/5~1/3 of the corresponding mechanical parameters of a rock block, and sometimes, the difference may be even greater, with the ratio reaching 1/20~1/10. The Poisson's ratio of a rock mass is generally 1.2~1.4 times greater than that of a rock block, and the joint stiffness is 0.1~0.9 times greater than that of a normal rock block. The rock mechanics parameters used in the numerical simulation were obtained via reduction based on the geological report of the Gushuyuan Mine and the experimental results of rock mechanics related to the #15 coal seam combined with the rock scale effect [26–28]. The specific parameters are shown in Table 1 below.

**Table 1.** Mechanical parameters of rock mass after reduction.

| Name of Rock Mass (GPa) | Bulk Modulus (Gpa) | Shear Modulus (MPa) | Tensile Strength (MPa) | Cohesion (MPa) | Angle of Internal Friction (°) | Density (Kg/m$^3$) |
|---|---|---|---|---|---|---|
| limestone | 24.32 | 12.50 | 1.82 | 4.23 | 38 | 2760 |
| 15# coal seam | 4.41 | 2.23 | 0.56 | 0.82 | 23 | 1540 |
| fine sandstone | 13.16 | 4.92 | 1.61 | 4.12 | 36 | 2628 |
| grit stone | 10.17 | 3.26 | 1.60 | 4.05 | 36 | 2672 |
| mudstone | 5.52 | 2.54 | 1.05 | 2.03 | 26 | 2570 |
| sandy mudstone | 6.87 | 2.93 | 3.35 | 1.18 | 28 | 2707 |

In order to study the influence of lateral abutment pressure on the stability of roadway surrounding rock, the influence of the working face advancing process on the surrounding

rock stress distribution is simulated using FLAC3D 6.0 software. The Mohr–Coulomb constitutive model is used in the numerical simulation. The normal displacement of the surface around the model is zero, and the normal and horizontal displacements at the bottom of the model are zero. The cloud map of the support pressure distribution of the working face advancing 200 m is obtained, as shown in Figure 5.

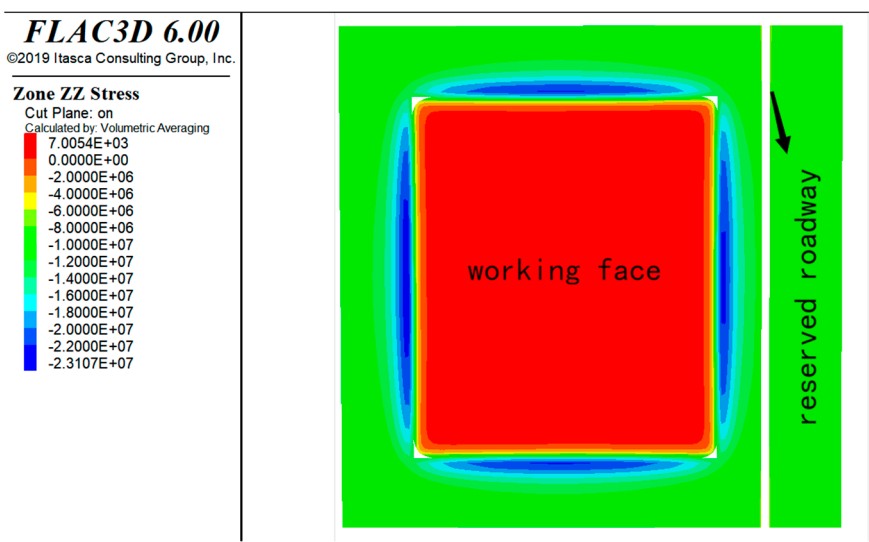

**Figure 5.** Working face advancing 200 m support pressure distribution cloud map.

In Figure 5, it can be seen that after the working face advances to 200 m, the areas of increased stress in the left and right coal bodies are large, the peak value of the abutment pressure reaches 20.5 MPa, the peak position is about 13 m from the coal wall of the goaf, the pressure relief area in the coal pillar is about 2 m, the area of increase stress is about 44 m, and the depth of the coal wall is 44 m. Outside of the original rock stress state, the coal pillar between the reserved roadway and the goaf in the #15 coal seam is only 25 m, and the excavation position is within the area of increased stress, as shown in Figure 6. The roadway is greatly affected by the abutment pressure of the working face during the mining period of the working face. After excavating the working face, the roof loses support in the vertical direction, resulting in the transfer of the roof load to the coal pillars on both sides, thus causing the deformation and failure of the 1523103 reserved roadway.

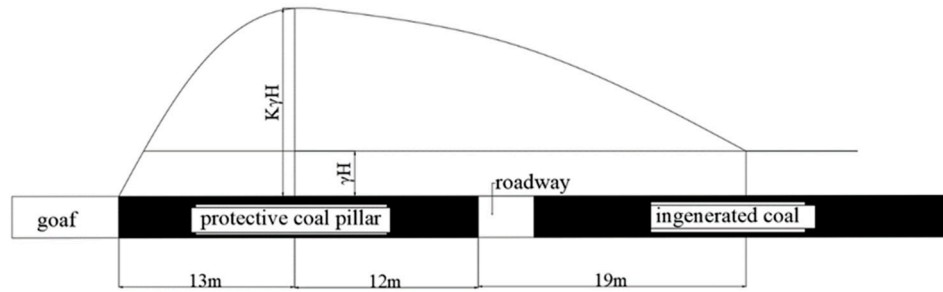

**Figure 6.** Relationship diagram of reserved roadway position and abutment pressure zone.

### 3.2. The Influence of Lateral Support Pressure on Working Face

After excavating the working face, the distribution of stress and the displacement of the roadway surrounding rock at different positions differ. Based on the upper section model, the FLAC3D 6.0 software was used to simulate the stress and displacement distribution of the surrounding rock at 25 m from the working face (stress increase area). The simulation results are shown in Figure 7.

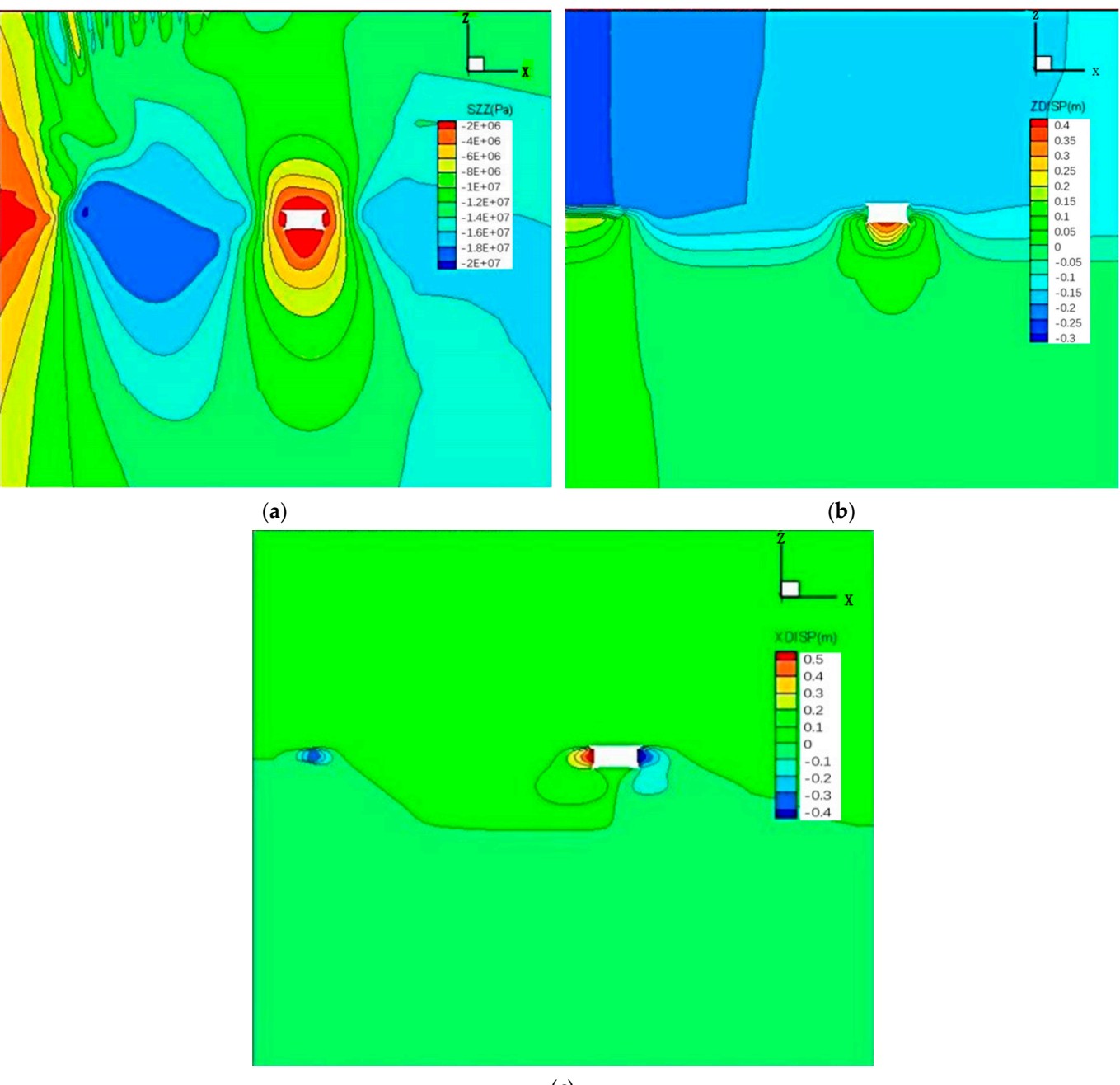

**Figure 7.** Cloud map of stress and displacement distribution in surrounding rock at a distance of 25 m from the working face (stress increasing area) in the roadway. (**a**) The vertical stress distribution cloud map of surrounding rock at 25 m (stress increase area) from the working face of roadway. (**b**) The vertical displacement cloud map of the roadway 25 m (stress increase area) from the working face. (**c**) The horizontal displacement cloud diagram of the roadway 25 m (where the stress increases) away from the working face.

It can be seen in Figure 7a that there is a large amount of vertical stress on the two sides of the roadway and the on roof and floor in the stress-increasing area, and the maximum vertical stress is about 2 MPa. The vertical stress of the two sides of the roadway changes violently from the surface to the depth, but the stress change in the roof and floor of the roadway from the surface to the depth is relatively gentle, which is due to the loss of vertical support in the roof after the excavation of the working face, resulting in the transfer of roof load to the coal pillars on both sides. The stress near the protective coal pillar is larger, and

the maximum stress can reach 20 MPa. It can be seen in Figure 7b that when the roadway is 25 m away from the working face (stress increase area), the roof subsidence is about 0.17 m, and the bottom displacement is about 0.30 m. The vertical displacement of the roadway roof is less than the vertical displacement of the floor. This is because the difference in the lithologic strength of the roof and floor of the roadway leads to inconsistency in the deformation of the roof and floor, and the roof of the roadway is thick and hard, which causes it to sink as a whole, and the subsidence is small. As can be seen from Figure 7c, the displacement of the protective coal pillar side is about 0.51 m, and the displacement of the solid coal side is about 0.48 m. The horizontal displacement of the roadway coal pillar side is larger than that of the solid coal side. This is because the abutment pressure in the protective coal pillar is greater than that in solid coal. The internal stress of a protective coal pillar is formed by the superposition of the working face lateral abutment pressure, and the stress concentration in the coal pillar is caused by roadway excavation. To sum up, the shrinkage rate of the roadway section is about 35.3%.

*3.3. Influence of Rock Mass Strength*

According to the experimental results of the physical and mechanical parameters of rock, the compressive strength of roof limestone is 92.41 MPa, the natural density is 2618 kg/m$^3$, the elastic modulus is 25.64 GPa, and the firmness coefficient is 9.2, which belongs to hard rock. It is difficult to completely collapse the hard roof during the mining process of the working face, but it is inclined to the goaf at a certain angle on the whole end face. The collapsed gangue has a certain supporting effect on the roof, and the direct roof forms a cantilever beam on the side of the protective coal pillar. The cantilever beam takes the coal wall as the fulcrum and transfers the pressure to the protective coal pillar while carrying the pressure of itself and the overlying strata in the goaf. As the pressure value continues to rise, when it is greater than the supporting strength of the protective coal pillar, the coal wall near the goaf begins to be destroyed, resulting in the transfer of stress to the inside of the coal pillar, which, in turn, increases the internal stress of the surrounding rock of the reserved roadway, resulting in roadway damage.

According to the field conditions, the calculation formula of the caving zone under hard conditions is selected [29–31].

$$Hm = \frac{100\sum M}{2.1\sum M + 16} \pm 2.5 \tag{1}$$

$\sum M$—cumulative mining thickness. The single-layer mining thickness is 1~3 m, and the cumulative mining thickness is not more than 15 m.

The thickness of limestone strata in the roof of the #15 coal seam is 10.13 m, and the mining height is 2 m; that is, the thickness of the roof strata is greater than three times the mining height. It can be seen that the height of the caving zone is 5.7 ~7.6 m, and the coefficient of bulking is generally 1.33~1.5. Combined with Formula (1), it is finally determined that the height of the roof caving zone of the #15 coal seam is 7 m~12 m.

The compressive strength of floor mudstone is 26.88 MPa, with a firmness coefficient of 4.7, and the compressive strength of coal seam is 22.68 MPa, with a firmness coefficient of 2.3. Due to the existence of clay minerals, the state and physical and chemical properties of mudstone are easily affected by water, pressure, temperature, and other environmental factors. It is easy to expand it in water, shrink it after water loss, and disintegrate it [32,33]. The two sides of the reserved roadway at 1523103 are mostly fractured bodies or even broken bodies. The mine pressure characteristics of the surrounding rock of the reserved roadway mainly depend on the occurrence and development of the deformation and failure of the two sides of the coal body. If the control is not good, it can easily cause the following sequence of events to occur: the two sides are squeezed and broken → the two sides experience spalling and collapse → the roof support of the two sides is weakened → floor heave is caused by the extrusion of the floor → the vicious cycle of the intensification of the damage of the two sides begins.

### 3.4. The Influence of Supporting Method

After the excavation of the roadway, the stress of the surrounding rock of the roadway will be redistributed; the stress state of the surrounding rock will change from three-dimensional to approximately two-dimensional; and the elastic zone, plastic zone, and broken zone will be generated in the surrounding rock of the roadway over time. The change in the "three zones" will gradually increase over time and eventually stabilize. According to the field geological parameters of the Gushuyuan Mine, the radius of the plastic softening zone, $R_p$ = 3.4 m, and the radius of fracture zone, $R_t$ = 2.74 m, can be calculated. Given the serious deformation and failure of the loose coal on the two sides of the roadway, some theoretical formulas from at home and abroad [34–36], such as the A. H. Wilson formula, large plate fracture theory formula, and limit equilibrium theory formula, can be used to calculate the failure depth of the two sides of the reserved roadway in combination with the parameters of the on-site geological conditions. It is known that the failure depth of the two sides is 4.54 m~5.96 m.

Roadway 1523103 only has three high-strength rebar bolts without longitudinal ribs with a diameter of 18 mm and a length of 2000 mm on each side of the reserved roadway, and the supporting strength can only control the deformation of the surrounding rock during the excavation of the reserved roadway, but it is far from meeting the deformation of the surrounding rock caused by the mining of the working face. During the mining process of the working face, the damage range of the surrounding rock of the roadway is further expanded, the length and strength of the original bolt cannot meet the requirements, and its supporting effect is lost.

## 4. Control Countermeasures of Roadway Stability

Based on the above analysis, the lateral support pressure produced by mining in the working face has an adverse impact on the roadway, and it is not easy to collapse the hard roof, which aggravates the mine pressure of the rock mass on both sides of the roadway and leads to the destruction of the rock mass and weak floor on both sides of the roadway, resulting in the failure of the original support. In order to effectively control the stability of the surrounding rock, the mine pressure of the roadway should be reduced, the stress condition of the two sides and floor should be improved, and the support of the rock mass and floor on both sides of the roadway should be strengthened. We decided to adopt the cooperative control technology of pressure relief, improving the mechanical properties of the surrounding rock and high-strength support to control the large deformation of the surrounding rock of the reserved roadway with a hard roof in the mine.

The blasting method was used to cut off the connection between the roadway roof and the goaf so as to reduce the transmission of mine pressure in the goaf and reduce the mine pressure of the roadway. The method of grouting was used to improve the mechanical properties of the rock mass on both sides of the roadway, improve its strength and bearing capacity, and enhance the anchorage performance of the surrounding rock. An anchor cable was used to strengthen the support of the roof and both sides of the roadway. The anchor cable can increase the support strength, expand the support range, and make the rock mass on both sides of the roadway form a larger range of a bearing carrier. The cooperative support technology can reduce the roof pressure of the roadway, improve the stress state of the surrounding rock, increase the bearing capacity of the surrounding rock, expand the bearing range of the rock mass on both sides of the roadway so as to reduce the destruction of the rock mass and floor on both sides of the roadway, and control the large deformation of the roadway.

## 5. Design of Surrounding Rock Control Scheme of Reserved Roadway 1523103

The main reasons for the deformation of the 1523103 reserved roadway include a large amount of lateral support pressure, the weak strength of the original support, the small strength of the rock mass on both sides of the roadway and floor rock mass, and the small strength of the supporting bolt. In view of the main causes of deformation, the surrounding



rock control scheme of roof cutting, the pre-splitting blasting of the roadway roof, and the grouting anchor cable providing high-strength support for the roadway roof and rock mass on both sides of the roadway are adopted to reduce the stress concentration in the coal pillars on both sides of the goaf and improve the bearing capacity of the rock mass and floor on both sides of the roadway so as to realize the long-term stability of the roadway surrounding rock.

*5.1. 152309 Face-Cutting Top Pressure Relief Technology*

5.1.1. Cutting Top Pressure Relief Principle

With regard to the #15 coal seam hard roof, due to its large thickness, high strength, good integrity, and being directly endowed in the upper part of the #15 coal seam, in the working face mining process, the hard roof will experience plastic bending with the advancement in the working face, showing the characteristic of slow sinking at the end of the working face to form a long cantilever beam and showing mine pressure through the cantilever beam to the coal column, thereby causing the roadway protection coal column to experience a high stress concentration and serious deformation damage. In this regard, the roof can be pre-cracked and blasted on the side of the coal column to protect the working face, the roof structure can be cut off between the 1523092 and 1523103 alleys, the length of the basic roof cantilever beam and the hanging area can be reduced, the roof energy storage conditions can be destroyed, the continuous force generated by the direct top rotation sinking inside the surrounding rock of the reserved roadway can be alleviated, the roof pressure along the empty side of the reserved roadway can be released, and the stress concentration degree of the lateral pressure in the surrounding rock of the roadway can be reduced. The stress distribution is shown in Figure 8.

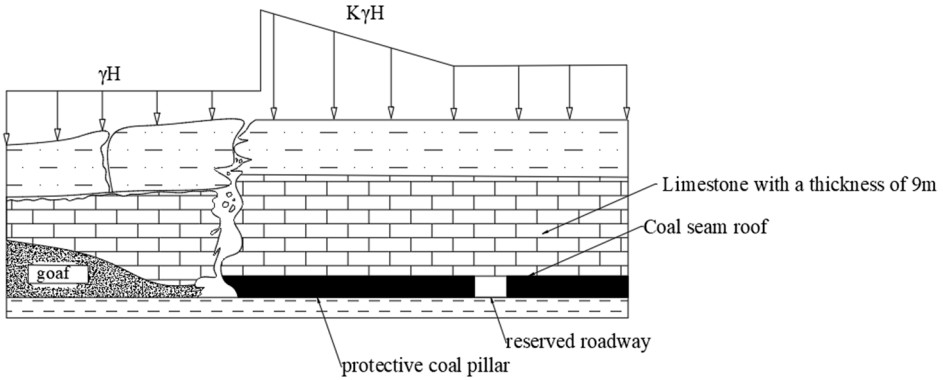

**Figure 8.** Schematic diagram of stress distribution after roof cutting of reserved roadway.

5.1.2. The Main Technical Parameters of Cutting Top Pressure Relief

According to the actual situation of the site and the workload situation, the end groove method is selected to weaken the top plate of the working face in order to reduce the initial pressure step distance of the working face. The average height of the working face is 1.7 m, so the effective blasting topping depth, H, can be calculated [37] using Equation (2) as 5.7 m, and 6 m is taken.

$$H = \frac{M}{K_P - 1} \tag{2}$$

M—height of mining; the value is 1.7 m.

$K_p$—volume expansion coefficient of rock after crushing; the value is 1.3.

Before the installation of the working face equipment, the roof is drilled 1 m away from the side of the old pond in the open-off cut. A total of 10 groups of blastholes are arranged in the 180 m long open-off cut: A, B, C, D, and E, and a, b, c, d, and e. Each group of A and a has two blastholes. B, C, and D and b, c, and d each have three holes. Both groups of E and e have four holes.

The specific parameters of A1 and a1 and A2 and a2 are as follows: the lengths of the blast holes are 14 m and 16 m, respectively, the elevation angles are 30° and 27°, respectively; the charge lengths are 6 m and 8 m, respectively; and the charge quantities are 15 kg and 20 kg, respectively.

The specific parameters of B1, b1; B2, b2; and B3, b3 are as follows: the hole lengths are 14 m, 14 m, and 16 m, respectively; the elevation angles are 32°, 30°, and 27°, respectively; the charge lengths are 7 m, 6 m, and 8 m, respectively; and the charge weights are 17.5 kg, 15 kg, and 20 kg, respectively.

The specific parameters of C1, c1; C2, c2; and C3, c3 are as follows: the hole lengths are 14 m, 14 m, and 16 m, respectively; the elevation angles are 32°, 30°, and 27°, respectively; the charge lengths are 7 m, 6 m, and 8 m, respectively; and the charge weights are 17.5 kg, 15 kg, and 20 kg, respectively.

The specific parameters of D1, d1; D2, d2; and D3, d3 are as follows: the hole lengths are 14 m, 14 m, and 16 m, respectively; the elevation angles are 32°, 30°, and 27°, respectively; the charge lengths are 7 m, 6 m, and 6 m, respectively; and the charge weights are 17.5 kg, 15 kg, and 20 kg, respectively.

The specific parameters of E1, e1; E2, e2; E3, e3; and E4, e4 are as follows: the lengths of the blast hole are 14 m, 14 m, 16 m, and 10 m, respectively; the elevation angles are 32°, 30°, 27°, and 30°, respectively; the charge lengths are 7 m, 6 m, 8 m, and 4 m, respectively; and the charge quantities are 17.5 kg, 15 kg, 20 kg, and 9.6 kg, respectively.

The remaining part of the blasthole is sealed with gun mud. Group A is symmetrical with group a, group B is symmetrical with group b, etc. The arrangement of the blasthole is shown in Figure 9.

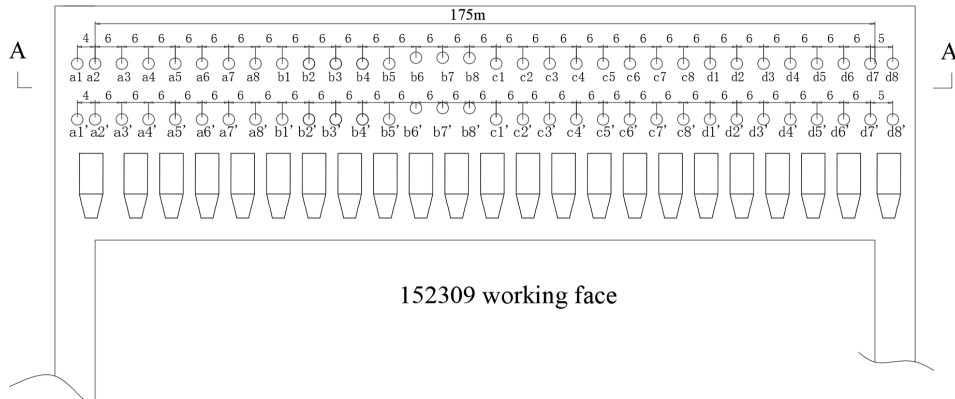

**Figure 9.** Blast hole layout plan.

*5.2. 1523092 Roadway Roof Weakening Technology*

5.2.1. Principle of Advanced Deep Hole Pre-Splitting Blasting Technology

Deep hole blasting can pre-split the thick and hard roof above the coal seam in advance, reduce the length of the main roof cantilever beam and the area of the suspended roof, and destroy the energy storage conditions of the roof. Deep hole blasting can loosen and pre-split the roof of the goaf in advance, release the roof pressure along the goaf side of the roadway, and effectively reduce the stress concentration degree of lateral pressure on the side of the protective coal pillar.

5.2.2. Main Technical Parameters of Advanced Deep Hole Pre-Splitting Blasting

A borehole with a length of 15 m and an elevation angle of 45° is arranged every 5 m from the cutting hole to the side of the coal pillar along the 1523092 roadway, with a horizontal angle of 0°, a charge length of 5 m, and a charge amount of 15 kg, and the remaining part of the borehole is filled with gun mud. Between the cut-through and the mining of the working face, 10 boreholes are arranged within 50 m of the cut-out to implement pre-splitting blasting. The specific blasthole parameters are shown in Table 2,

and the blasthole parameters can be adjusted appropriately according to the on-site blasting effect. During the blasting, two rows of single pillars are arranged every 1 m from the current working face at 0.6 m from the two sides of the blasting roadway to be cracked, which is used to strengthen the advance support after roof cutting and pre-splitting and ensure the stability of the 1523092 roadway during pre-splitting blasting. With the mining of the working face, such a reciprocating cycle always maintains a 50 m advance support during blasting. The spatial relationship between the advance support and the roof cutting hole is shown in Figure 10.

**Table 2.** Pre-splitting blasting hole parameter table.

| Hole Length (m) | Angle of Elevation (°) | Horizontal Angle (°) | Aperture (mm) | Charge Length (m) | Powder Charge (Kg) | Sealing Length (m) |
| --- | --- | --- | --- | --- | --- | --- |
| 15 | 45 | 0 | 75 | 5 | 15 | 10 |

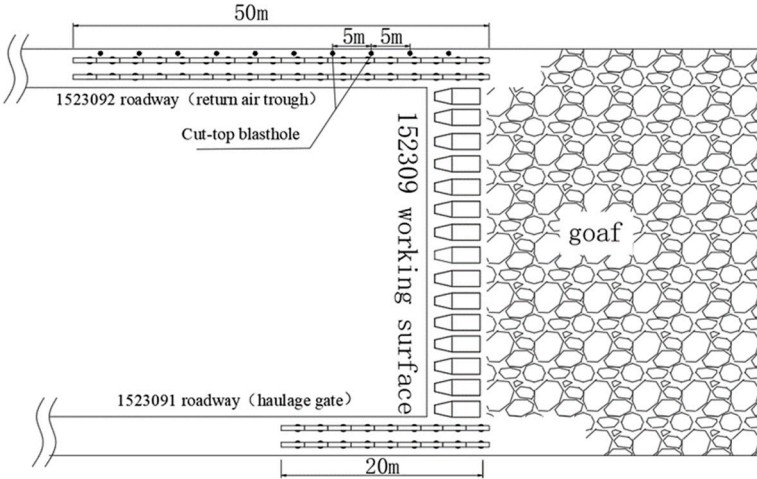

**Figure 10.** The spatial relationship diagram of advance support and roof cutting blasthole.

### 5.3. Grouting Anchor Cable Support Technology

5.3.1. Supporting Principle of Grouting Anchor Cable

The principle of grouting anchor cable reinforcement is to combine the supporting effect of the anchor cable with the effect of grouting reinforcement and work together on the surrounding rock of the roadway. The grouting of organic or inorganic slurry using a grouting anchor cable can not only fundamentally ensure the reliability of the anchor cable, but can also penetrate a large range of coal and rock mass around the drilling hole, produce a bonding and curing effect on the loose coal and rock mass, significantly improve its integrity, improve the self-supporting ability of the coal and rock mass, and greatly improve the supporting effect of the roadway. The structure diagram of the grouting anchor cable is shown in Figure 11.

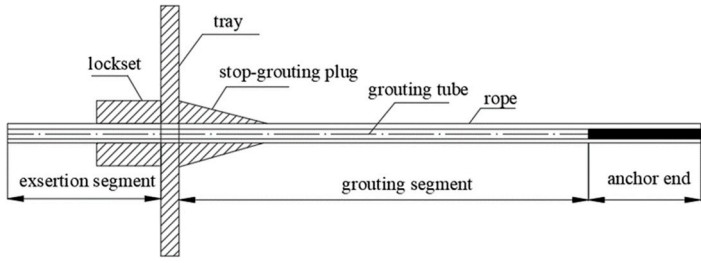

**Figure 11.** Structure diagram of grouting anchor cable.

### 5.3.2. Layout Scheme and Main Technical Parameters of Grouting Anchor Cable

On the basis of the original support of the 1523103 roadway, the grouting anchor cable is added to the coal wall from the two sides of the reserved roadway parallel to the floor to strengthen the control of the surrounding rock—that is, the hollow grouting anchor cable with dimensions of Φ22 mm × 5300 mm is added to the three-hole staggered type between the existing rows of anchor bolts and the upper and lower rows of the anchor cables are added with Φ14 mm trapezoidal steel strips, respectively. The grouting anchor cable layout is shown in Figure 12.

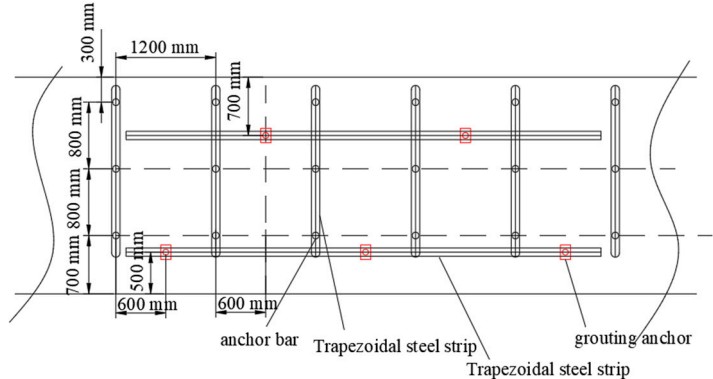

**Figure 12.** Arrangement diagram of grouting anchor cable.

According to the principle of the anchor cable length, strength, and key parts, the anchor cable row spacing is set to 1.2 m, the nominal diameter of the steel wire is 6.0 mm, the installation aperture is Φ32 mm, the strength is 1760 MPa, the breaking force is ≥420 KN, the resin anchorage length is 1000~1500 mm, the inner diameter of the hollow grouting pipe is Φ7.5 mm and the outer diameter is Φ10 mm, and the grouting pressure is 5.0~7.0 MPa. The new support scheme is shown in Figure 13.

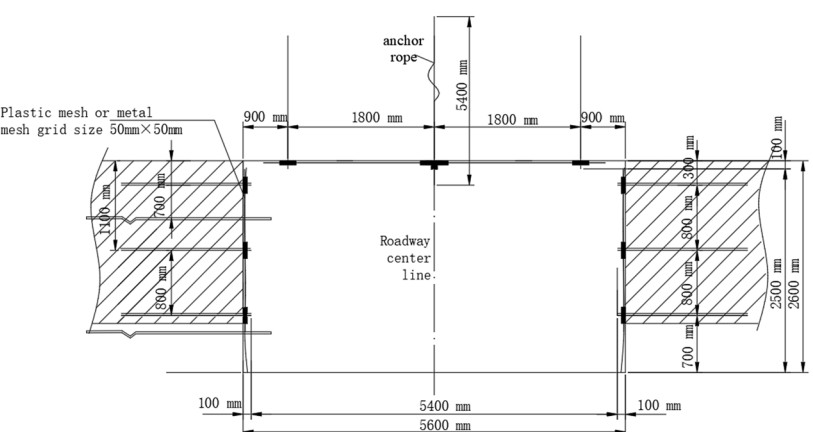

**Figure 13.** New support scheme diagram.

## 6. Analysis of Monitoring Results of Surrounding Rock Control in Reserved Roadway

### 6.1. Analysis of Monitoring Results of Lateral Support Pressure of Coal Pillar

Two survey areas were arranged outward along the contact lane at the open-off cut of the 1523103 roadway, and three boreholes were arranged in each survey area. In the #1 survey area, the #6 measuring point was 785 m away from the open-off cut, the #5 measuring point was 5 m away from the #6 measuring point, the #4 measuring point was 5 m away from the #5 measuring point, and the drilling installation depths were 5 m, 9.5 m, and 7 m, respectively. In the #3 survey area, the #4 measuring point was 900 m away from the open-off cut, the #2 measuring point was 5 m away from the #3 measuring point, the #1

measuring point was 5 m away from the #2 measuring point, and the drilling installation depths were 5 m, 6.5 m, and 8 m, respectively. Each borehole was 1 m higher than the floor and parallel to the floor of the coal pillar. The layout of the borehole stress meter is shown in Figure 14.

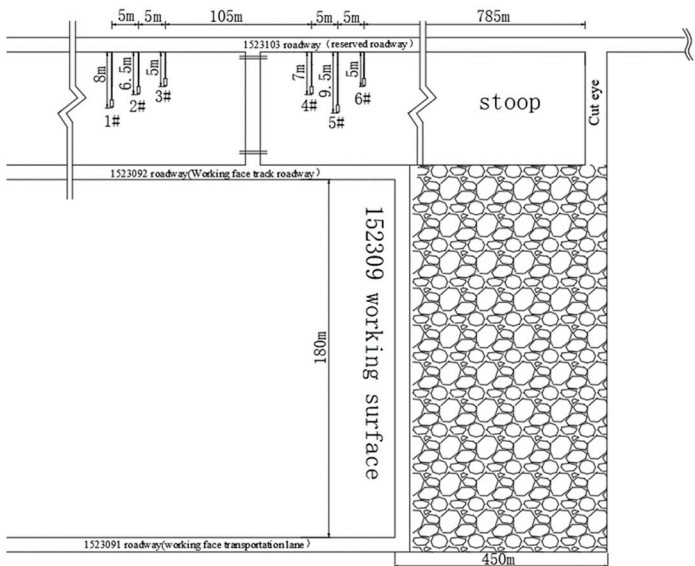

**Figure 14.** Borehole stress meter layout diagram.

According to the above design, six stress meters were arranged in the 1523103 roadway, and the placement depth of the stress meter was 7 m. Among them, the #5 stress gauge was destroyed by underground workers, and the #2 and #4 stress gauges had no pressure data after liquid injection, which was regarded as invalid, so there were three effective stress gauges. The stress curve is shown in Figure 15.

It can be seen in Figure 15 that there is no obvious change in the pressure value of the #1 borehole stress meter 50 m outside of the working face, and there is a decreasing trend between 9 m and 50 m in front of the working face. When the 1# drilling stress gauge is located 9 m in front of the working face, the stress value increases rapidly. When it is located 78 m behind the working face, the data basically do not increase. When the #1 borehole stress meter is located 9 m in front of the working face and 16 m behind the working face, its increment reaches 47% of the initial pressure. When the #1 borehole stress meter is located 16 m behind the working face, its increment only reaches 5% of the initial pressure. When the #1 borehole stress meter is located 78 m behind the working face, its value basically does not change. As can be seen from Figure 7a, the maximum stress at the drilling position before the top cutting is about 16 MPa; from Figure 15a, it can be seen that the maximum stress at the drilling position after the top cutting is only 4.2 MPa, which is about 73% lower than that without the top cutting.

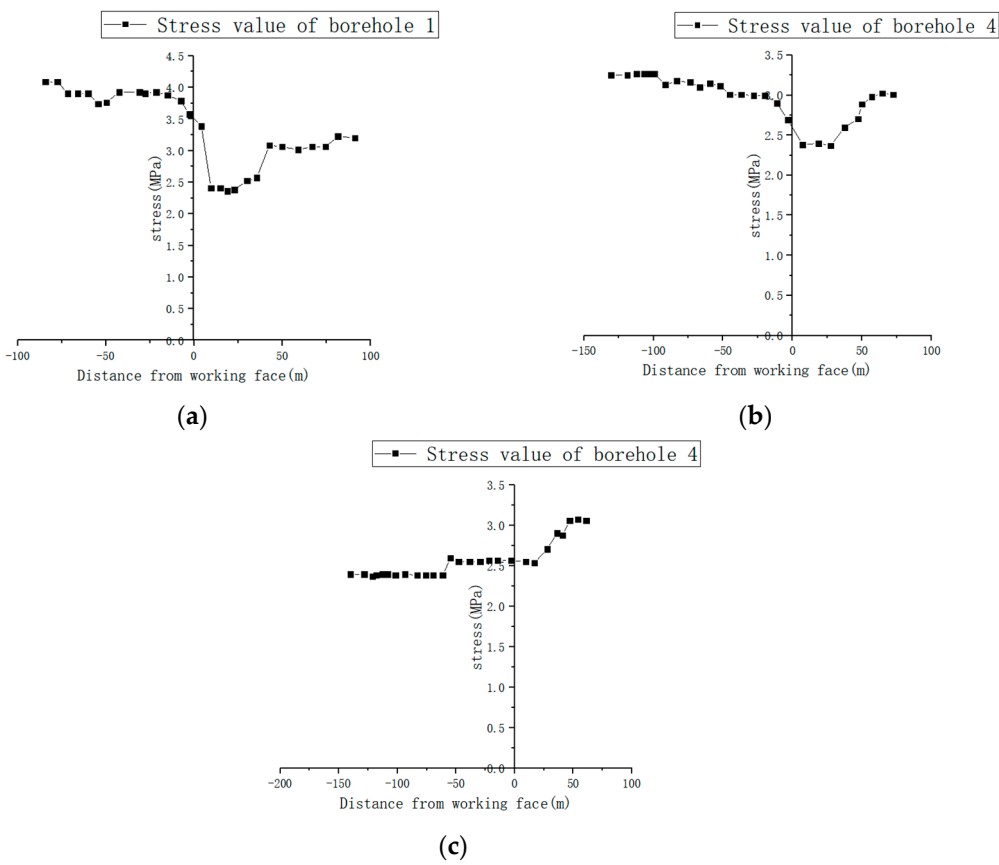

**Figure 15.** Stress curve. (**a**) Pressure curve of #1 borehole stress meter. (**b**) Pressure curve of #4 borehole stress meter. (**c**) Pressure curve of #6 borehole stress meter.

There is no obvious change in the pressure value of the #4 borehole stress meter 50 m outside of the working face, and there is a decreasing trend between 6 m and 40 m in front of the working face. When it is located in front of the working face at 6 m, the rate of increase is obvious. When it is located 90 m behind the working face, the data basically do not increase. When the #4 borehole stress meter is located in front of the working face at 6 m and 20 m behind the working face, the increment reaches 20% of the initial pressure. When the #4 borehole stress meter is located 20 m behind the working face, the increment only reaches 8% of the initial pressure. When the #4 borehole stress meter is located 90 m behind the working face, its value is basically unchanged. As can be seen from Figure 7a, the maximum stress at the drilling position before the top cutting is about 16 MPa; from Figure 15a, it can be seen that the maximum stress at the drilling position after the top cutting is only 3.3 MPa, which is about 79% lower than that without the top cutting.

The pressure value of the #6 borehole stress meter gradually decreases outside the working face at 26 m. When it is located at the working face of 26 m, the pressure meter value begins to stabilize and then basically does not change, indicating that the 1523103 roadway is 780 m away from the cut. The mining influence of the working face is very small, and there is no obvious significant influence range in this section. As can be seen from Figure 7a, the maximum stress at the drilling position before the top cutting is about 16 MPa; from Figure 15a, it can be seen that the maximum stress at the drilling position after the top cutting is only 3.2 MPa, which is about 80% lower than that without the top cutting.

Compared with before and after the top cutting of the working face, the lateral support pressure of the coal pillar is reduced by at least 73% and up to 80%, and the effect of roof cutting and pressure relief is obvious. In general, the influence range of the 1523103 roadway affected by mining is concentrated between 9 m in front of the working face and 90 m behind the working face, and the significant influence range is concentrated between

9 m in front of the working face and 20 m behind the working face. Even so, the relative value of the lateral coal wall support pressure of the reserved roadway is still very small, and the maximum is 4.2 MPa, that is, the mining of the working face has no obvious influence on the stability of the reserved roadway.

*6.2. Analysis of Surface Displacement Monitoring Results of Reserved Roadway*

In the 1523103 reserved roadway, six measuring points (#1~#6) are arranged from the inside to the outside of the open-off cut position. The #1 measuring point is 785 m away from the open-off cut, the #2 measuring point is 790 m away from the open-off cut, the #3 measuring point is 795 m away from the open-off cut, the #4 measuring point is 800 m away from the open-off cut, the #5 measuring point is 900 m away from the open-off cut, and the #6 measuring point is 910 m away from the open-off cut. The positions of #1, #2, #3, #5, and #6 correspond to those of the #6, #5, #4, #3, and #1 stress meters, respectively.

The surface displacement of the 1523103 reserved roadway was observed using a steel ruler and measuring rod. According to the data obtained from measuring points #1~#6, the surface displacement deformation observation results of each measuring point were drawn, as shown in Figure 16.

In the diagram, it can be seen that the maximum shrinkage of the two sides of the #1 measuring point is 44 mm, the maximum shrinkage of the two sides of the #2 measuring point is 23 mm, the deformation of the two sides of the #3, #4, #5, and #6 measuring points is within 15 mm, and the deformation curve is gentle. In general, there is almost no deformation in the two sides of the roadway.

When the #1 measuring point is located 33 m in front of the working face, the convergence of the roof and floor begins to increase greatly. When the measuring point is located 108 m behind the working face, the convergence of the roof and floor gradually tends to be gentle. The maximum shrinkage of the roof and floor of the #1 measuring point is 51 mm. The change trend of the roof and floor convergence of the #2, #3, and #4 measuring points is similar to that of the #1 measuring point. The maximum shrinkage of the roof and floor of the #2 measuring point is 23 mm, the maximum shrinkage of the roof and floor of the #3 measuring point is 40 mm, and the maximum shrinkage of the roof and floor of the #4 measuring point is 49 mm. The roof-to-floor convergence of the #5 and #6 measuring points is large. When the #5 measuring point is located 24 m in front of the working face, the roof-to-floor convergence increases greatly, and the maximum shrinkage of the roof-to-floor convergence is 91 mm. When the #6 measuring point is located 34 m in front of the working face, the convergence of the roof and floor increases greatly, and the maximum shrinkage of the roof and floor is 101 mm.

Based on the above analysis, the deformation pattern of the 1523103 reserved tunnel during the period affected by the mining face is roughly 29.5 m away from the leading working face as the dividing line. The roadway beyond 29.5 m of the advanced working face is basically not affected by the mining of the working face, the two sides of the roadway and the roof and floor of the roadway have no obvious deformation, and the roadway basically maintains its original stable state. The roadway within 29.5 m of the leading working face began to deform under the pressure of the leading support of the working face, and the approaching amount of the roof and floor began to increase rapidly. Through observation and data analysis, the deformation of the reserved roadway is mainly bottom heave, and the effect of controlling the surrounding rock deformation of the reserved roadway is better after adopting the new surrounding rock control scheme, as shown in Figure 17.

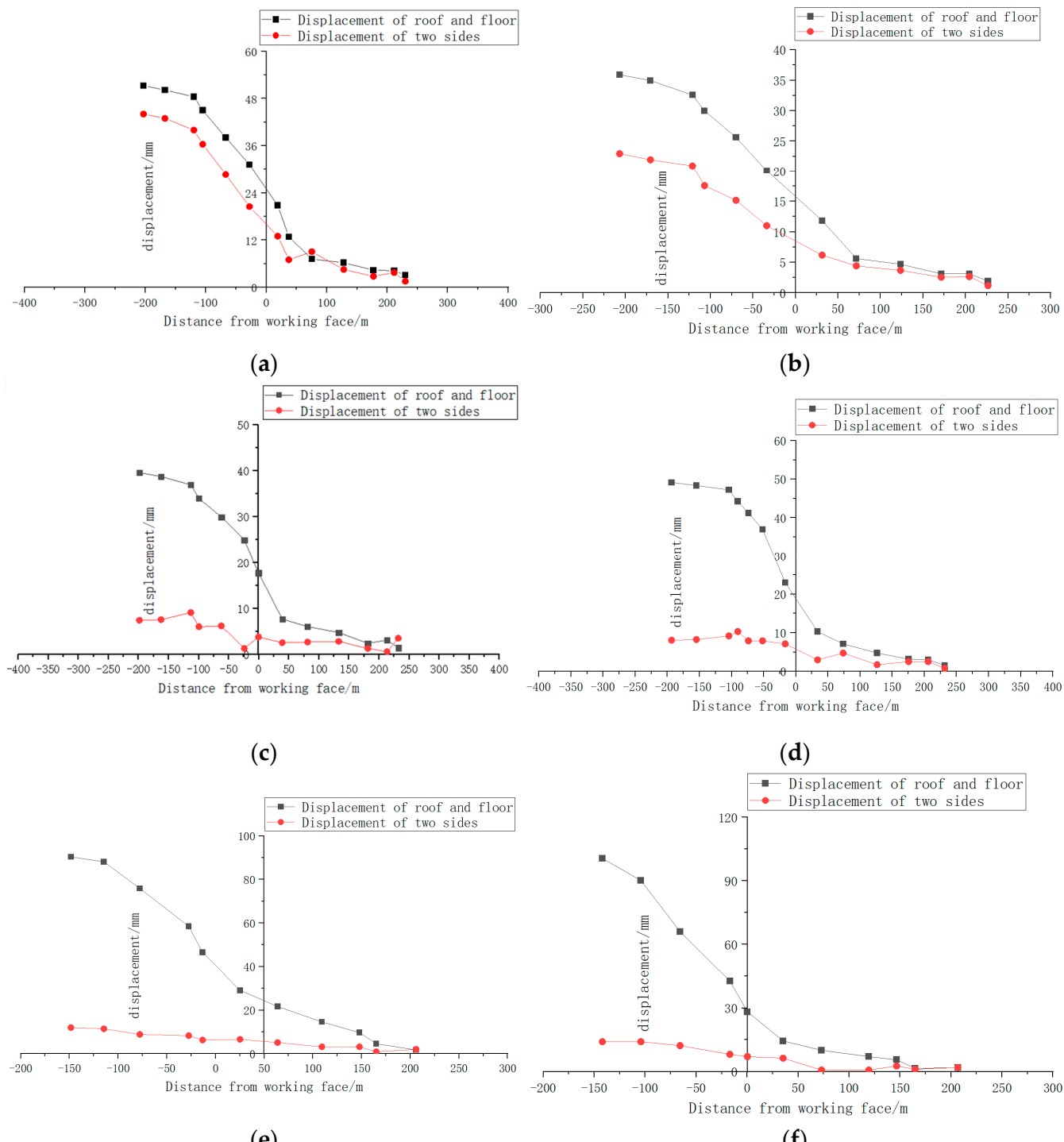

**Figure 16.** Surface displacement curve of measuring point. (**a**) Surface displacement curve of #1 measuring point. (**b**) Surface displacement curve of #2 measuring point. (**c**) Surface displacement curve of #3 measuring point. (**d**) Surface displacement curve of #4 measuring point. (**e**) Surface displacement curve of #5 measuring point. (**f**) Surface displacement curve of #6 measuring point.

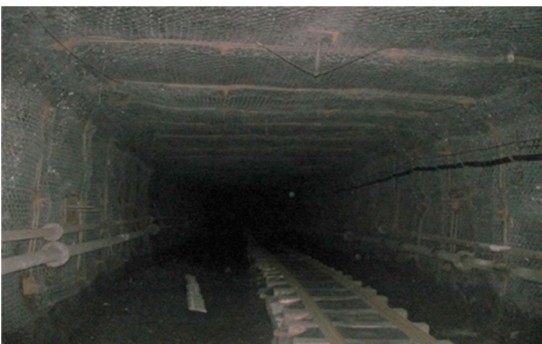

**Figure 17.** Real picture of deformation control effect of reserved roadway surrounding rock.

## 7. Conclusions

The 1523103 roadway is greatly affected by the supporting pressure of the working face during mining. After the working face is excavated, the roof load is transferred to the coal pillars on both sides, causing fragments of the 1523103 reserved roadway. The floor of the 1523103 roadway is mudstone, and its own state and physical and chemical properties are easily affected by environmental factors such as water, pressure, temperature, etc. It easily swells when it encounters water, it shrinks after losing water, and it is prone to floor heaving. The roadway floor heave reduces the roadway section, restricts the coal mine transportation work, hinders the mine ventilation, consumes a lot of labor and material resources, seriously limits the output efficiency of the coal mine, and is not conducive to the sustainable development of the coal mine.

The cooperative control technology involving relieving pressure and improving the mechanical properties of the surrounding rock and high-strength support is adopted to control the large deformation of surrounding rock of the reserved roadway with a hard roof in the mine, reduce the roof pressure of the roadway, improve the stress state of the surrounding rock, increase the bearing capacity of the surrounding rock, and expand the bearing range of the rock mass on both sides of the roadway so as to reduce the destructive effect of both sides of the roadway and floor. After adopting the surrounding rock control scheme of roof cutting and pressure relief + bolt + grouting anchor cable support, the maximum bottom heave of the reserved roadway is reduced from 1.5 m to 0.1 m, and the maximum shrinkage of the cross section is reduced from 70% to 11%. The maximum deformation of the two sides of the roadway is 0.044 m, the deformation curve is smooth, and the surrounding rock control effect is good.

The successful treatment of the large deformation of the surrounding rock of the reserved roadway with a hard roof by cutting the top pressure + anchor rod + grouting anchor cable support reduces the number of protective coal pillars, improves the coal mining rate, ensures the safety of miners, prolongs the service life of coal mines, and is conducive to the sustainable utilization of non-renewable resources and the sustainable development of the coal industry.

**Author Contributions:** All the authors contributed to this paper. W.W. conceived of and designed the research; Y.H. and M.L. analyzed the data and wrote the manuscript; Z.L. and Z.Z. provided theoretical and methodological guidance in the research process; M.L. participated in revising the manuscript. All authors have read and agreed to the published version of the manuscript.

**Funding:** This research was funded by the Cultivation of National Major Achievements, grant number NSFRF230202; the Henan Province University Science and Technology Innovation Talent Support Program, grant number 23HASTIT011; and the Innovative Research Team of Henan Polytechnic University, grant number T2022-2.

**Institutional Review Board Statement:** Not applicable.

**Informed Consent Statement:** Not applicable.

**Data Availability Statement:** Not applicable.

**Conflicts of Interest:** The authors declare no conflict of interest.

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
