# Peer review of "Study on the Stress Distribution and Stability Control of Surrounding Rock of Reserved Roadway with Hard Roof"

_sustainability, doi:10.3390/su151914111_

Round 1

Reviewer 1 Report

I carefully reviewed the research material. The article has scientific and practical value, is logical and understandable. However, in the process of reading the article, I came up with a few recommendations, solely to improve the quality of the article.

1. In my opinion, the analysis carried out in the introduction looks insufficient. Only 11 scientific works of scientists were analyzed. Show the importance of the problem of stability of mine workings in general. Briefly give known methods for achieving the stability of mine workings (anchor fastening, backfilling of goaf, explosive destruction of the roof).

2. Before the last paragraph of the introduction, I would recommend the authors to formulate a few specific proposals, what has not been studied enough today? What problem has not been solved, what specifically has not been paid attention by scientists. This will show the originality of the presented research.

3. Part of the article within lines 148-185 refers to research methodology. Here, the authors present a research mechanism using numerical simulation. It is logical to call this part of the article "Justification of the geomechanical model ...".

4. After line 185, I would recommend that you make a new subsection 3.2 and name it as subsection 3.1 is called. in the authors' version of the article.

5. In lines 200-215, where the authors give an analysis of the stress distribution, I would like to see a quantitative expression of the results. By what percentage or how many times does the voltage decrease/increase?. This is an important scientific result.

6. In my opinion, it is necessary to improve the quality of Figures 7-9. The voltage scale is not visible.

7. In Figure 11, show the symbol for the top of the coal seam.

8. How are the hole parameters justified in Table 2? Probably there should be a connection with the results of the numerical simulations outlined above.

9. It would be appropriate to add a photo of the mine measurements in section 5.

10. I have carefully analyzed the list of references and I have a wish for the authors to also cite scientists from other countries. Your bibliography consists of 27 scholarly sources from China. However, the problems of stability of underground mine workings in the zone of influence of the treatment works are dealt with by a significant number of scientists from different countries.

I would recommend to consider several scientific papers in the field of mine workings sustainability and supplement the list of references:

Malashkevych, D., Petlovanyi, M., Sai, K., Khalymendyk, O. (2022). Influence of rock leaving in the longwall face goaf on the extraction drift stability

ARPN Journal of Engineering and Applied Sciences, 17(21), 1924-1934.

Shavarskyi, I., Falshtynskyi, V., Dychkovskyi, R., Akimov, O., Sala, D., & Buketov, V. (2022). Management of the longwall face advance on the stress-strain state of rock mass. Mining of Mineral Deposits, 16(3), 78–85. https://doi.org/10.33271/mining16.03.078

Also a positive aspect would be the addition of 2-3 scientific papers by scientists from other countries.

Author Response

请参阅附件。

Reviewer 2 Report

The paper discusses field observation and analysis that attribute the failure of reserved roadway 1523103 to lateral support pressure, rock strength, and support mode. Increased pressure from adjacent coal pillars due to mining at working face 152309 leads to spalling and floor heave. Numerical simulation using FLAC3D software revealed stress and displacement patterns post-roof cutting and pressure relief. A control approach involving roof cutting, pressure relief, and grouting anchor cable support is discussed. 

See attached document for more detailed comments.

Reviewer 3 Report

The manuscript (entitled: Study on the stress distribution and deformation control of surrounding rock after roof cutting and pressure relief in re-served roadway with a hard roof) pays main attention on a combined control technology proposed based on roof cutting and pressure relief and supplemented by grouting anchor cable support considering the causes of surrounding rock failure in reserved roadways. This topic is important and the work is in detail. Yet, I still think the manuscript should be improved in the content. My detail comments are listed as following points.

1) The title is too long and the authors should shorten it to highlight the representative topic.

2) The abstract should be revised to highlight the innovative work.

3) The figures 7,8 and 9 is not clear and should be revised.

4) How to decide the mechanical model and get the inputted parameters for numerical simulation?

5) How about the initial geostress condition in your numerical simulation?

6) How to simulate the plastic mesh and anchor rope is not clear.

7) Can you add a comparing curves analysis between the numerical data and measured stress and deformation?

8) In the conclusion, can the authors summarize the practical key technologies for deformation control for the surrounding rock after roof cutting and pressure relief in re-served roadway with a hard roof?

Reviewer 4 Report

Based on the site conditions of the mine mining face, this article obtained the main reasons for the occurrence of sidewalls and floor heaves in the reserved roadway through on-site investigation and theoretical analysis. Numerical simulation was carried out using FLAC3D finite element software, and the hard roof reserved roadway was obtained. Stress distribution and displacement of surrounding rock after roof cutting and pressure relief. Aiming at the failure reasons of the surrounding rock of the reserved roadway, the author proposes a combined control technology that mainly focuses on roof cutting and pressure relief, supplemented by grouting and anchor cable support, and the field application effect is good. I think the surrounding rock control technology proposed in this article is practical and innovative, and it is a good article. Based on the content of the article, the following five suggestions are proposed.

1. The abstract of the article needs to be further refined.

2. In Chapter 3.2, the rock mass properties of the roof and floor of the roadway should be further enriched, and the physical properties of the hard rock should be specified in detail, because the physical properties of the hard rock have an important impact on the collapse degree of the roof of the roadway.

3. In the last paragraph of Chapter 5.2, a comparative analysis should be made to more clearly explain the deformation of the reserved roadway.

4. The presentation of the conclusion section needs to be further streamlined.

5. The language in the article needs further polishing and improvement.

Round 2

Reviewer 1 Report

I have carefully read the updated version of the article.

The authors took into account the recommendations and improved the article.

I recommend the article for publication. As a reviewer, everything suits me.

Good luck to the authors in further research.

Sincerely,

reviewer

Author Response

We appreciate for your warm work earnestly, thank you very much for your comments and suggestions.

Reviewer 3 Report

This revision is acceptable.

Author Response

(The authors gave the same response as above.)
